# Learning Graph Neural Networks with Noisy Labels

**Hoang NT***, **Choong Jun Jin & Tsuyoshi Murata**
Department of Computer Science
Tokyo Institute of Technology
{hoangnt,junjin.choong}@net.c.titech.ac.jp, murata@c.titech.ac.jp

## Abstract

We study the robustness to symmetric label noise of GNNs training procedures. By combining the nonlinear neural message-passing models (e.g. Graph Isomorphism Networks, GraphSAGE, etc.) with loss correction methods, we present a noise-tolerant approach for the graph classification task. Our experiments show that test accuracy can be improved under the artificial symmetric noisy setting.

## 1 Introduction

Large datasets are beneficial to modern machine learning models, especially neural networks. Many studies have shown that the accuracy of machine learning models grows log-linear to the amount of training data (Zhou, 2017). Currently, complex machine learning models can only achieve super-human classification results when trained with a very large dataset. However, large datasets are usually expensive to collect and create exact label. One solution to create large datasets is crowd-sourcing, but this approach introduces a higher level of labeling error into the datasets as well as requires a lot of human resources (Georgakopoulos et al., 2016). As a consequence, neural networks are prone to very high generalization error under noisy label data. Figure 1 demonstrate the accuracy results of a graph neural network trained on MUTAG dataset. Training accuracies tend to remain high while testing accuracies degrades as more label noise is added to the training data.

Graph neural network (GNN) is a new class of neural networks which learn from graph-structured data. Typically, GNNs classify graph vertices or the whole graph itself. Given the input as the graph structure and data (e.g. feature vectors) on each vertex, GNNs training aim to learn a predictive model for classification. This new class of neural networks enables end-to-end learning from a wider range of data format. In order to build large scale GNNs, it requires large and clean datasets. Since graph data is arguably harder to label than image data both at vertex-level or graph-level, graph neural networks should have a mechanism to adapt to training label error or noise.

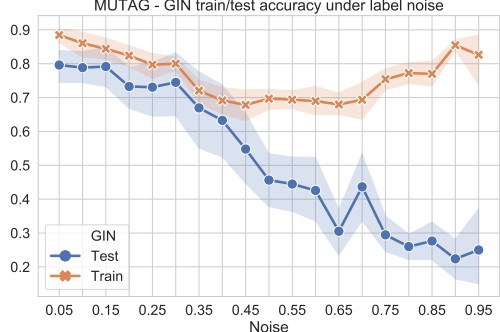

Figure 1: GIN model trained with increasing symmetric label noise. The generalization gap increases as more noise is introduced to the training labels.

In this paper, we take the noise-correction approach to train a graph neural network with noisy labels. We study two state-of-the-art graph neural network models: *Graph Isomorphism Network* (Xu et al., 2019) and *GraphSAGE* (Hamilton et al., 2017). Both of these models are trained under symmetric artificial label noise and tested on uncorrupted testing data. We then apply label noise estimation and loss correction techniques (Patrini et al., 2016; 2017) to propose our denoising graph neural network model (D-GNN).

---

*https://github.com/gear/denoising-gnn

## 2 METHOD

### 2.1 GRAPH NEURAL NETWORKS

**Notations and Assumption** Let $\mathcal{G} = (V, E, \boldsymbol{X})$ be a graph with vertex set $V$, edge set $E$ and vertex feature vector matrix $\boldsymbol{X} \in \mathbb{R}^{|V| \times f}$, where $f$ is the dimensionality of vertex features. Our task is graph classification with noisy labels. Given a set of graphs: $\{\mathcal{G}_1, \mathcal{G}_2, \ldots, \mathcal{G}_N\}$, their labels $\{\tilde{y}_1, \tilde{y}_2, \ldots, \tilde{y}_N\} \subset 2^m$, we aim to learn a neural network model for graph label prediction: $y_\mathcal{G} = f(\mathcal{G})$. We assume that the training data is corrupted by a noise process $\boldsymbol{N}$, $\mathrm{N}_{i,j}$ is the probability label $i$ being corrupted to label $j$. We further assume $\boldsymbol{N}$ is symmetric, which corresponds to the symmetric label noise setting. Noise matrix $\boldsymbol{N}$ is unknown, so we estimate $\boldsymbol{N}$ by learning correction matrix $\boldsymbol{C}$ from the noisy training data.

**GNN Models** The most modern approach to the graph classification problem is to learn a graph-level feature vector $\mathbf{h}_\mathcal{G}$. There are several ways to learn $\mathbf{h}_\mathcal{G}$. *GCN* approach by Kipf & Welling (2017) approximates the Fourier transformation of signals (feature vectors) on graphs to learn representations of a special vertex to use as the representative for the graph. Similar approaches can be founded in the context of compressive sensing. To overcome the disadvantages of GCN-like methods such as memory consumption and scalability, the nonlinear neural message passing method is proposed. *GraphSAGE* (Hamilton et al., 2017) proposes an algorithm consists of two operations: `aggregate` and `pooling`. `aggregate` step computes the information on each vertex using the local neighborhood, then `pooling` computes the output for each vertex. These vector outputs are then used in classification at vertex-level or graph-level. More recently, *GIN* (Xu et al., 2019) model generalizes the concept in *GraphSAGE* to propose a unified message-passing framework for graph classification.

### 2.2 LEARNING NOISY LABEL DATA

**Surrogate Loss** Using an alternative loss function to deal with noisy label data is a common practice in the weakly supervised learning literature (Natarajan et al., 2013; Biggio et al., 2012; Georgakopoulos et al., 2016; Patrini et al., 2016; 2017). We apply the *backward* loss correction procedure to graph neural network: $\ell^{\leftarrow} = \boldsymbol{C}^{-1} \cdot \ell(\hat{p}(y|\mathcal{G}))$. This loss can be intuitively understood as *going backward one step* in the noise process $\boldsymbol{C}$ (Patrini et al., 2017).

We study the *symmetric* noise setting where label $i$ is corrupted to label $j$ with the same probability for $j$ to $i$ ($\mathrm{N}_{i,j} = \mathrm{N}_{j,i}$) (Biggio et al., 2012). We use a $m \times m$ symmetric Markov matrix $\boldsymbol{N}$ to describe the noisy process with $m$ labels. Furthermore, to simplify the experiment settings, with a given $n$ we set: $\mathrm{N}_{i,j} = \mathrm{N}_{i,k} = n \; \forall j, k \neq i$. For example when $m = 3, n = 0.2$ the noise matrix is:

$$\boldsymbol{N} = \begin{bmatrix} 0.8 & 0.1 & 0.1 \\ 0.1 & 0.8 & 0.1 \\ 0.1 & 0.1 & 0.8 \end{bmatrix}$$

Matrix $N$ above can be interpreted as all labels are *kept* with probability $0.8$ and *corrupted* to other labels with probability $0.2$ (summation of off-diagonal elements in a row).

### 2.3 DENOISING GRAPH NEURAL NETWORKS

Formaly we define our graph neural network model as the message passing approach proposed by Xu et al. (2019). The feature vector $\mathbf{h}_v$ of a vertex $V$ at $k$-th hop (or layer) is given by AGGREGATE and COMBINE functions:

$$\begin{aligned} \mathbf{a}_v^{(k)} &= \text{AGGREGATE}^{(k)}(\{\mathbf{h}_u^{(k-1)} : u \in \mathcal{N}(v)\}), \\ \mathbf{h}_v^{(k)} &= \text{COMBINE}^{(k)}(\mathbf{h}_v^{(k-1)}, \mathbf{a}_v^{(k)}) \end{aligned} \tag{1}$$

$\mathcal{N}(v)$ denotes the neighborhood set of vertex $v$; and $k \in [K]$ is the predefined number of "layers" corresponding to network's perceptive field. The final representation of graph $\mathcal{G}$ is calculated using a READOUT function. Then, we train the neural network by optimizing the surrogate backward loss.

$$\mathbf{h}_{\mathcal{G}} = \text{READOUT}(\{\mathbf{h}_v^{(K)} : v \in \mathcal{G}\}),$$
$$\ell^{\leftarrow}(p(y|\mathbf{h}_{\mathcal{G}}), y_{\mathcal{G}}) = \boldsymbol{C}^{-1} \cdot \text{CROSS\_ENTROPY}(p(y|\mathbf{h}_{\mathcal{G}}), y_{\mathcal{G}}) \tag{2}$$

D-GNN is different from *GIN* only at the surrogate loss function as described above. To train a D-GNN model, we first train a *GIN* model on the noisy data for estimating $\boldsymbol{C}$, then we train D-GNN using the estimated correction matrix.

We train our D-GNN model using three different noise estimator: Conservative (D-GNN-C), Anchors (D-GNN-A), and Exact (D-GNN-E). The exact loss correction is introduced for comparison purposes. The hyperparameters of our models are set similar to GIN model in the previous paragraph. For conservative and anchor correction matrix estimation, we train two models on the same noisy dataset: The first model is without loss correction and the second model is trained using the correction matrix from the first model. For all neural network models, we use the ReLU activation unit as the nonlinearity.

## 3 EMPIRICAL RESULTS

We test our framework on the set of well-studied 9 datasets for the graph classification task: 4 bioinformatics datasets (MUTAG, PTC, NCI1, PROTEINS), and 5 social network datasets (COLLAB, IMDB-BINARY, IMDB-MULTI, REDDIT-BINARY, REDDIT-MULTI5K) (Yanardag & Vishwanathan, 2015). We follow the preprocessing suggested by Xu et al. (2019) to use one-hot encoding as vertex degrees for social networks (except REDDIT datasets). Table 1 gives the overview of each dataset. Since these datasets have exact label for each graph, we introduce symmetric label noise artificially.

### 3.1 NOISE ESTIMATION

Table 1: Data overview

| Dataset | #graphs | #classes | #vertices |
|---------|---------|----------|-----------|
| IMDB-B | 1000 | 2 | 19.8 |
| IMDB-M | 1500 | 3 | 13.0 |
| RDT-B | 2000 | 2 | 429.6 |
| RDT-M5K | 5000 | 5 | 508.5 |
| COLLAB | 5000 | 3 | 74.5 |
| MUTAG | 188 | 2 | 17.9 |
| PROTEINS | 1113 | 2 | 39.1 |
| PTC | 344 | 2 | 25.5 |
| NCI1 | 4110 | 2 | 29.8 |

**Conservative Estimation** We estimate the corruption probability by the Conservative Estimator described in the previous sections. For each noise configuration, we train the original neural network (GIN) on the noisy data and use the neural response to fill each row of the correction matrix $\boldsymbol{C}$. Table 2 gives an overview of how well the conservative estimation matrix diverges from the correct noise matrix. The matrix norm $\|\boldsymbol{C} - \boldsymbol{N}\|$ is the $p$-norm with $p = 1$.

**Anchor Estimation** We follow the noise estimation method introduced in Patrini et al. (2017) (Equations (12,13)) to estimate the noise probability using an unseen set of samples. These anchor samples are assumed to have the correct labels, hence they can be used to estimate the noise matrix according to the expressivity assumption. In our experiments, these samples are taken from the testing data (one per class). Table 2 demonstrates the similarity results.

Table 2: Norm distance between conservative correction matrix estimation $\boldsymbol{C}^c$ and $\boldsymbol{C}^a$ compared with true noise matrix $\boldsymbol{N}$ when $n = 0.2$

| Dataset (#classes) | diag($N$) | Avg. diag($\boldsymbol{C}^c$) | $\|\boldsymbol{C}^c - N\|$ | Avg. diag($\boldsymbol{C}^a$) | $\|\boldsymbol{C}^a - N\|$ |
|--------------------|-----------|-------------------------------|----------------------------|-------------------------------|----------------------------|
| IMDB-B (2) | 0.8 | 0.99 | 0.76 | 0.77 | 0.12 |
| IMDB-M (3) | 0.8 | 0.99 | 1.14 | 0.85 | 0.30 |
| RDT-B (2) | 0.8 | 0.99 | 0.76 | 0.75 | 0.20 |
| RDT-M5K (5) | 0.8 | 0.99 | 1.90 | 0.81 | 0.10 |
| COLLAB (3) | 0.8 | 0.99 | 1.14 | 0.75 | 0.30 |
| MUTAG (2) | 0.8 | 0.99 | 0.76 | 0.74 | 0.24 |
| PROTEINS (2) | 0.8 | 0.99 | 0.76 | 0.78 | 0.08 |
| PTC (2) | 0.8 | 0.99 | 0.76 | 0.63 | 0.68 |
| NCI1 (2) | 0.8 | 0.99 | 0.76 | 0.74 | 0.24 |

**Exact Assumption**    In this experiment setting, we assume that the noise matrix is exactly known from some other estimation process. In practice, such an assumption might not be realistic. However, under the symmetric noise assumption, the diagonal of the correction matrix $C$ can be tuned as a hyperparameter.

## 3.2    GRAPH CLASSIFICATION

We compare our model with the original Graph Isomorphism Network (GIN) (Xu et al., 2019). The hyperparameters are fixed across all datasets as follow: `epochs=20`, `num_layers=5`, `num_mlp_layers=2`, `batch_size=64`. We keep these hyperparameters fixed for all datasets since the similar trend of accuracy degradation is observed independently of hyperparameter tuning. Besides GIN, we consider GraphSAGE model (Hamilton et al., 2017) under the same noisy setting. We use the default setting for GraphSAGE as suggested in the original paper.

Table 3: Classification results at symmetric noise, when $n = 0.2$ (80% data has correct labels). We calculate the mean and std of accuracy score on test data for 10 runs each configuration. Bold font indicates improvement compared to the original model.

|  | MUTAG | IMDB-M | RDT-B | RDT-M5K | COLLAB | IMDB-B | PROTEINS | PTC | NCI1 |
|---|---|---|---|---|---|---|---|---|---|
| GIN | .7327 | .4476 | .6695 | .3677 | .6544 | .6573 | .6257 | .4824 | .6472 |
| GraphSAGE | .7072 | .4373 | - | - | - | .6410 | .6583 | .4892 | .6053 |
| D-GNN-C | .5727 | **.4747** | .5005 | .2000 | .5979 | **.6940** | **.6693** | **.5557** | .6170 |
| D-GNN-A | .7102 | **.4505** | .5307 | .2000 | **.6917** | **.7088** | **.6769** | **.5001** | .6405 |
| D-GNN-E | .7002 | **.4633** | .5270 | .2022 | **.6960** | **.7190** | **.6917** | **.5235** | **.6638** |

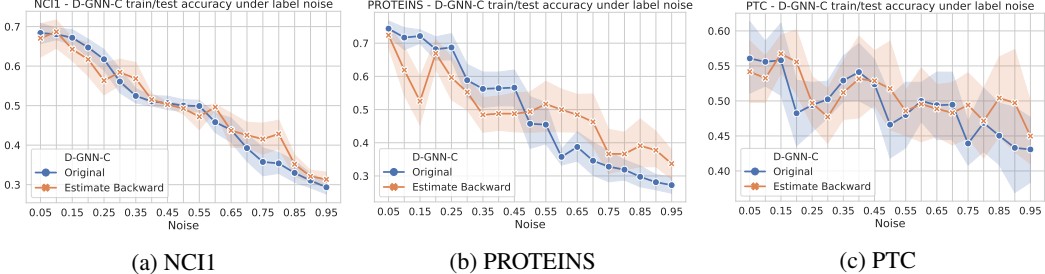

|(a) NCI1|(b) PROTEINS|(c) PTC|

Figure 2: Denoising results on bioinformatics datasets. X-axis presents the test accuracies.

We fix the noise rate at 20% for the experiments in Table 3 and report the mean accuracy after 10 fold cross validation run. The worst performance variance of our model is the conservative estimation model. Due to the overestimation of softmax unit within the cross-entropy loss, the model's confidence to all training data is close to 1.0. Such overconfidence leads to wrong correction matrix estimation, which in turn leads to worse performance (Table 2). In contrast to D-GNN-C, D-GNN-A and D-GNN-E have consistently outperformed the original model. Such improvement comes from the fact that the correction matrix $C$ is correctly approximated. Figure 2 suggests that the D-GNN-C model might work well under the higher label noise settings.

## 4    CONCLUSION

In this paper, we have introduced the use of loss correction for Graph Neural Networks to deal with symmetric graph label noise. We experimented on two different practical noise estimatation methods and compare them to the case when we know the exact noise matrix. Our empirical results show some improvement on noise tolerant when the correction matrix $C$ is correctly estimated. In practice, we can consider $C$ as a hyperparameter and tune it following some clean validation data.

ACKNOWLEDGMENTS

This work was supported by JSPS Grant-in-Aid for Scientific Research (B) (Grant Number 17H01785) and JST CREST (Grant Number JPMJCR1687).

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
