# OpenReview forum: "Learning Graph Neural Networks with Noisy Labels"
_ICLR.cc/2019/Workshop/LLD — LLD 2019_

### Official Review · AnonReviewer2 · 2019-04-07
**Interesting work but need clear writing**

**Rating:** 3
**Confidence:** 2

**Review:**

Summary: This paper introduces the loss correction for Graph Neural Networks (GNN) to deal with symmetric graph label noise. The paper shows interesting works on GNN, but the writing could be clearer to deliver the proposed idea.

Notes:
- The paper shows an interesting work on graph datasets. It has the potential for diverse graph-related tasks.
- The paper focuses on a graph classification task. It would be better to show performances on a node classification task.
- One area which is not clear is the justification of the worst performance of the conservative estimation model. What about the original models? GIN and GraphSAGE also use cross-entropy loss, but they show much better performances than D-CNN-C, A, and E in some datasets. (What is the 'original model'? GIN?)
- The writing could be clearer to deliver the proposed idea. Explanations for notations are missing in some parts. Also, the paper needs clearer definitions of each model.

The paper introduces an interesting denoising approach on GNN, and the proposed model shows good performances on datasets. There are some unclear areas in the paper, which should be addressed before final submission.

---

### Official Review · AnonReviewer1 · 2019-04-08
**Paper which tackles an interesting problem, but still needs to be improved.**

**Rating:** 2
**Confidence:** 2

**Review:**

This work proposes the use of a noise correction loss in the context of graph neural networks to deal with noisy labels. The GNN is implemented following the message passing approach proposed by Xu (2019). The authors compare 3 different noise estimators (namely the conservative, anchors and exact approaches) in the task of graph classification using 9 datasets.

The paper tackles an interesting and relevant problem to the community. The contribution of the proposed loss in real settings is not clear since only experiments with synthetic noise were performed. More importantly, in its current form the paper is not easy to follow and there are missing details and omissions that should be corrected:

- Until we read the “Empirical results” section, it is not clear what are the differences between the conservative, anchor and exact methods. The description given in the “Empirical results” section should be part of the “Method” section.

- For the conservative approach, it is not clear what is the loss function used to train the first model which estimates C.

- In Sec 2.1, what is "m" in 2^m ? Is this the number of possible labels?

- What is C^a in Table 2?

- Table 3 indicates “We calculate the mean and std of accuracy score on test data for 10 runs each configuration”. However, there is only one value reported in the table which I assume corresponds to the mean value.

- Figure 2 indicates “X-axis presents the test accuracies”. As far as I understand, test accuracies are indicated in the Y-axis.

---

### Decision · Program_Chairs · 2019-04-16
**Acceptance Decision**

Accept